# Molecular Analysis of Anti-Tuberculosis Drug Resistance of *Mycobacterium tuberculosis* Isolated in the Republic of Korea

**DOI:** 10.3390/antibiotics12081324

**Published:** 2023-08-17

**Authors:** Se-Mi Jeon, Sanghee Park, Na-Ra Lim, Noori Lee, Jihee Jung, Nackmoon Sung, Seonghan Kim

**Affiliations:** 1Division of Bacterial Disease Research, Korea National Institute of Health, Korea Disease Control and Prevention Agency, Cheongju-si 28159, Republic of Korea; sem21c@korea.kr (S.-M.J.); park3hee@korea.kr (S.P.); jo77311@korea.kr (N.-R.L.); 2Clinical Research Center, Masan National Tuberculosis Hospital, Changwon-si 51755, Republic of Korea; newry@korea.kr (N.L.); loody23@gmail.com (J.J.); paratb@mf.seegene.com (N.S.)

**Keywords:** *Mycobacterium tuberculosis*, drug resistance, DST, 24-locus MIRU-VNTR, spoligotyping

## Abstract

Rapid and accurate detection of tuberculosis (TB) drug resistance is critical for the successful treatment and control of TB. Here, we investigated resistance to anti-TB drugs and genetic variations in 215 drug-resistant *Mycobacterium tuberculosis* isolates in Korea. Genetic variations were observed in *rpoB* Ser531Leu, *katG* Ser315Thr, and *gyrA* Asp94Gly; however, the minimum inhibitory concentrations varied, which can be attributed to other resistance mechanisms. Examination of genetic relatedness among drug-resistant isolates revealed that the cluster size of resistant bacteria was less than six strains, suggesting no evidence of a large-scale epidemic caused by a specific strain. However, *rpoC* mutants of the rifampicin-resistant isolates were composed of five types of clusters, suggesting that these compensatory mutations advance propagation. In the present study, more than 90% of the resistance mechanisms to major anti-TB drugs were identified, and the effect of each mutation on drug resistance was estimated. With the clinical application of recent next-generation sequencing-based susceptibility testing, the present study is expected to improve the clinical utilization of genotype-based drug susceptibility testing for the diagnosis and treatment of patients with drug-resistant TB.

## 1. Introduction

A major public health challenge associated with tuberculosis (TB) is the emergence of multidrug-resistant TB (MDR-TB) and extensive drug-resistant TB (XDR-TB). Treatment of MDR-TB is associated with a longer treatment period, higher cost, and more adverse drug events than drug-sensitive TB, while XDR-TB is more difficult to treat and has a relatively higher mortality rate than MDR-TB [1,2].

Patients with drug-susceptible TB undergo an initial six-month treatment utilizing first-line drugs, namely rifampicin (RIF), isoniazid (INH), ethambutol (EMB), and pyrazinamide (PZA). If they become resistant to RIF and/or INH, they are treated for an extended period with second-line drugs, such as fluoroquinolones (FQ) and aminoglycosides (AG). RIF interferes with the transcription process and kills *Mycobacterium tuberculosis*. *rpoB*, encoding the beta subunit of RNA polymerase, is involved in the resistance to RIF [3,4]. Moreover, the degree of resistance varies according to the position of the mutation. Globally, the highest *rpoB* mutation rate is observed in Ser531 (56.19 ± 9.14%), followed by His526 (17.81 ± 6.33%) and Asp516 (12.94 ± 6.95%) [5,6,7,8,9,10,11]. In particular, in the case of Ser531, more than 90% of single mutations are associated with high resistance to RIF [12], and more than 85% of Ser531 and His526 mutations are simultaneously involved in resistance to rifabutin. Mutations in *rpoB* may inhibit the growth of resistant bacteria; the phenomenon is referred to as the ‘fitness cost’ effect, which can be compensated for by mutations in *rpoA* or *rpoC* [13,14,15,16,17,18,19,20,21].

Although the mechanism of action of INH is not understood comprehensively, it is generally known to inhibit cell wall synthesis and various processes in cells. Genes related to resistance to INH, *katG*, *inhA*, *ahpC*, and others, have been reported [22,23,24,25]. Mutations related to INH resistance have been demonstrated to vary based on the country of identification, with a distribution of 47.7–91.7% for *katG* Ser315, followed by *inhA* C(-15)T mutations, at 7.0–35.5%. The *katG* Ser315 mutation is also considered to affect fitness cost [13,26]. In addition, the *katG* mutation is mainly related to high INH tolerance, and the *inhA* mutation is associated with low tolerance if it is a single mutation.

Another primary drug, EMB, interferes with cell wall composition and kills bacteria. The *embCAB* operon is associated with resistance to EMB, and mutations most related to resistance have occurred in *embB* [27,28,29].

Patients resistant to the primary drugs RIF and INH are subsequently administered secondary drugs, specifically AG and FQ, for their treatment. AG-based drugs used this way include streptomycin (STR), amikacin (AMK), kanamycin (KAN), and capreomycin (CAP), and related resistance genes include *rrs*, *eis*, and *rpsL* [30,31,32,33,34,35,36,37,38].

FQ targets enzymes essential for the survival and proliferation of bacteria, such as DNA gyrase and topoisomerase IV. Resistance to the drugs is most often due to mutations in *gyrA*, *gyrB*, *parC*, and *parE* that encode targets of the drug. However, resistance to FQ in *M. tuberculosis* primarily arises from mutations in *gyrA* and *gyrB*, as the bacterium lacks topoisomerase IV [39,40,41,42,43,44,45,46,47].

In Korea, the number of new patients with TB has been decreasing continuously since 2012, and the number of patients with MDR-TB had decreased to approximately one-third (399) by 2020 compared with that in 2012 (1212). The percentage of patients with MDR-TB in Korea is approximately 1.57%, which is lower than the global average (3.4%); however, caution should be exercised considering the increasing trend (1.3–7.7%) of MDR-TB among foreign patients in Korea [48].

Molecular DST determines drug resistance based on mutations in resistance-associated genes. Massive data are required to understand the impact of such resistance mutations on susceptibility to specific anti-TB drugs. Recently, the World Health Organization (WHO) announced recommendations on direct drug susceptibility testing (DST) using next-generation sequencing (NGS), indicating the increasing utilization of molecular DST [49,50,51]. To facilitate the exploration of the correlation between *M. tuberculosis* complex mutations and drug resistance, DST’s PPV (positive predictive value of mutation) was established based on the resistance gene mutations identified in several studies. Based on the catalog, the susceptibility and resistance of anti-tuberculosis drugs can be predicted according to the gene mutation location. In the present study, the distribution of genetic variations related to drug resistance was investigated, and the levels of resistance among TB strains isolated in Korea were compared based on the types of resistance-related mutations. To assess the possibility of an outbreak of a particular drug-resistant TB strain, the genetic relatedness of TB strains isolated in Korea was investigated.

## 2. Results

### 2.1. Anti-TB DST

Among the 215 tested isolates, 190 were resistant to at least one anti-TB drug, and 25 were susceptible to all tested anti-TB drugs. Sixty-four isolates corresponded to MDR-TB, and seventy were pre-extensively drug-resistant (pre-XDR)-TB, the phase prior to XDR. Isoniazid (INH)-resistant strains were the most common, with 155 isolates, followed by 152 RIF-resistant and 95 ofloxacin (OFX)-resistant strains.

### 2.2. RIF Resistance

Table 1 represents the minimum inhibitory concentrations (MICs) and DNA sequencing results of RIF and *rpoB* against 195 *M. tuberculosis* isolates. One hundred and fifty-one isolates had point mutations concentrated in ten different codons; among them, one isolate also had an insertion, whereas deletion was only identified in one isolate. Single mutations were found in 145 isolates, and 7 isolates showed double mutations.

The most common mutation, Ser531Leu, was identified in 85 isolates. Irrespective of the presence of other mutations, the Ser531Leu mutation showed an MIC > 16 µg/mL for RIF.

The second most common mutation occurred in codons 516 (27 isolates) and 526 (24 isolates). Codons 516 and 526 showed four and eight different amino acid changes, respectively. Among these, 13 isolates harbored the Asp516Val mutation, which is commonly observed after the Ser531Leu mutation.

Seven isolates had Leu533Pro, a representative disputed mutation, and exhibited an MIC of 1–2 µg/mL for RIF. Nine isolates had another disputed mutation, Asp516Tyr, five of which showed a low MIC (≤2 µg/mL), and four of which showed a high MIC (≥8 µg/mL) for RIF in the absence of accompanying mutations. Another disputed mutation, His526Leu, found in three isolates, showed an MIC of ≥8 µg/mL for RIF.

### 2.3. INH Resistance

The nucleotide sequences of *katG*, *inhA*, and *ahpC* were analyzed, and the MIC for INH was compared in 199 isolates in association with the mutations (Table 2).

In 15 INH-resistant isolates, among a total of 153, no mutations were detected in *katG*, *inhA*, and *ahpC*. However, 19 susceptible isolates harbored mutations in *katG* and *inhA*.

Mutations in *katG* were identified in 111 isolates, including two susceptible strains. Among them, 16 *katG* mutants harbored mutations in *inhA*.

Variations in *katG* showed 14 types of mutations in 13 codons, and 97 isolates of *katG* mutants had the Ser315Thr mutation. Most Ser315Thr-associated MICs for INH were higher than 2 µg/mL, but five isolates showed an MIC ranging from 0.5 to 1 µg/mL.

Mutations in the *inhA* promoter were identified in five types at four positions. Of these, the C-to-T change at position -15 of the *inhA* promoter was the most frequent, with 54 isolates harboring this mutation. Among the C(-15)T mutants, eight occurred simultaneously with the Ser315Thr mutation of *katG* in the isolates.

In addition, the *ahpC* mutation was confirmed in nine isolates, and seven mutation types were found at six positions. One isolate harbored additional mutations in *katG*.

### 2.4. Ethambutol (EMB) Resistance

A total of 116 strains were examined for mutations in *embB*, and the MIC for EMB was calculated (Table 3). An *embB* mutation was identified in 91 isolates, 24 of which were susceptible. Among the resistant isolates, seven had no mutations.

Nineteen different types of mutations were found in 11 codons. In addition, four isolates had double mutations, and eighty-seven had a single mutation.

The most frequent mutations were substitutions in the codon of Met306 of *embB*. We identified 57 isolates with a mutation in the codon; 30 isolates had a Met306Val mutation, 26 isolates had a Met306Ile mutation, and 1 isolate had a Met306Leu mutation. The MIC_50_ of the isolate differed depending on the type of mutation present. The MIC_50_ values of isolates with Met-to-Val mutation and Met-to-Ile mutation were 8 µg/mL and 4 µg/mL, respectively. Twelve isolates with a Met306Ile mutation were in the susceptible range for EMB (4 µg/mL).

### 2.5. Fluoroquinolone (FQ) Resistance

Based on the MIC of OFX, mutations in *gyrA* and *gyrB* were investigated in 170 isolates (75 OFX-susceptible and 95 OFX-resistant strains), and the MIC values of all strains were compared (Table 4). Mutations in codons 21 and/or 95 of *gyrA* were excluded from the analysis, as they have been reported to be natural polymorphisms that are not related to FQ resistance.

Mutations in *gyrA* were concentrated in four codons in 81 isolates. Mutations in *gyrB* occurred in seven codons of 26 isolates.

Sixty-three isolates had only a single mutation, and three had double mutations within *gyrA*. Eleven isolates had a single mutation in *gyrB*, fourteen isolates had single mutations in both *gyrA* and *gyrB*, and one isolate had a three-point mutation: one in *gyrA* and two in *gyrB* (Table 5).

The most common mutation occurred in *gyrA* at codon 94 Asp in 54 isolates, and all strains were transformed into five different codons; among these, codon 94 Asp-to-Gly mutants were the most prevalent, with 31 isolates. The Ala90Val mutation, observed in 21 isolates of *gyrA* mutants, was the second most frequent variation. The most common mutation in *gyrB* was Asp500Asn, which was detected in five isolates.

Even with a single mutation in *gyrA*, the MIC for OFX was 8 µg/mL or higher. When a single mutation was present in *gyrB*, the MIC for OFX was lower (4 µg/mL or below). When simultaneous mutations were present in *gyrA* and *gyrB*, the MIC was significantly higher (16 µg/mL or higher).

### 2.6. Aminoglycoside (AG) Resistance

The sequence of *rrs* was analyzed in 125 isolates, and among them, the *eis* gene sequence was examined in 89 isolates. The sequencing results and MIC for amikacin (AMK) and kanamycin (KAN) were compared. Of the 125 isolates analyzed, 87 were susceptible, and 38 were resistant to KAN (Table 6).

Sequencing of *rrs* revealed mutations in 35 out of the 125 isolates. Mutations were identified in seven codons with single mutations, except for one isolate with a double mutation. The most prevalent mutation was the A1401G substitution in *rrs*, and the 24 isolates with the mutation induced a strong resistance of at least 40 µg/mL to KAN and 16 µg/mL to AMK. However, mutations in other positions, such as positions 514 and 517, either did not affect drug susceptibility or resulted in a slight decrease in sensitivity.

Mutations in the *eis* promoter were identified at three positions in 10 isolates. The mutations result in decreased sensitivity or low levels of resistance to KAN at its MIC. However, for AMK, most mutations showed an MIC of ≤1 µg/mL, which did not affect the sensitivity.

The *rpsL* sequence was analyzed in 118 isolates containing 59 streptomycin (STR)-resistant strains and compared with the sequencing results of the previously analyzed *rrs* (Table 7). Mutations were found in 38 isolates, all of which were single mutations identified in five types at three codon positions. Among them, 38 isolates were within the resistance range, and 1 isolate showed an MIC, indicating susceptibility for STR but near the critical concentration. The most common mutation of *rpsL*, Lys43Arg, occurred in 28 isolates. Regardless of the presence of another mutation, all Lys43Arg mutations induced an MIC of >32 µg/mL for STR. Two isolates, mutated to methionine at the same position, were not accompanied by mutations of other genes and had an MIC of >32 µg/mL for STR. Lys88Arg, the second most common mutation, was not accompanied by other site mutations and induced an MIC range of 8 to 32 µg/mL.

The correlation between AG resistance and mutations of the common *rrs* was that the *rrs* A1401G mutation induced resistance to KM and AK but not to STR. Conversely, mutations in bases 514, 517, and 908 induced the resistance to STR but did not affect resistance to KM and AK.

### 2.7. Compensatory Mutation

Mutations in *rpoC* were identified in 35 out of the 181 isolates. Mutations were identified as 20 types in the 17 codons. Of the 35 isolates, 1 isolate had a double mutation, and the remaining isolates had a single mutation. In addition, all *rpoC* mutants harbored mutations in *rpoB*. The mutation types in the *rpoB* were Ser531Leu, with 33 isolates, and His526Asp and His526Arg in the remaining isolates.

The most prevalent mutation was Phe452Leu, with six isolates harboring this mutation. The second was Pro434Val, which was detected in five isolates. Codon 491 isoleucine was mutated into two types: three isolates had a substitution for threonine, and the other three were substituted with valine. The Gly388Ala and Lys445Arg mutants contained His526Asp and His526Arg mutations in *rpoB*, respectively.

An examination of the genetic association of *rpoC* mutants using 24 locus mycobacterial interspersed repetitive unit–variable number of tandem repeats (MIRU-VNTR) of the 35 strains in which *rpoC* was mutated revealed that 14 isolates formed five clusters with a size of 2–5 strains. Four isolates of *rpoC* Phe452Leu mutants composed a cluster, and all five Phe452Leu-mutant isolates formed their own clusters. All three isolates of Ile491Thr mutants also composed clusters (Figure 1).

### 2.8. Genetic Relevance of Resistant Strains

Genotypic lineages were identified by spoligotyping 190 isolates that were resistant to at least one of the anti-TB drugs investigated in the present study (Appendix A). A total of 162 isolates (85.3%) were classified as the East Asian lineage (lineage 2), including the Beijing family, and 21 isolates belonged to the Euro-American lineage (lineage 4).

Genetic association analysis of the resistant strains using 24- locus MIRU-VNTR identified 163 types, of which 16 formed cluster sizes consisting of two to six isolates, while the remaining 143 isolates had unique types.

It was difficult to observe a correlation between each MIRU-VNTR type with anti-tuberculosis drug susceptibility pattern or region and time of isolation. However, there was a correlation with the mutation type of the *rpoC* gene (Figure 1).

## 3. Discussion

In the present study, we investigated variations in genes related to anti-TB drug resistance in isolates from Korea. The results of this study provide evidence that could improve the accuracy of anti-TB resistance determination using molecular DST. It is vital to know the extent to which a mutation affects anti-TB resistance, in addition to determining whether the mutation is in a related gene. Therefore, the anti-TB susceptibility levels of the isolates used in the present study were determined using the MIC test, and the gene sequence related to the resistance to each anti-TB agent was analyzed to confirm its association with the MIC.

The WHO recently revised the definition of XDR-TB, according to which pre-XDR-TB refers to TB that meets the criteria for XDR-TB and shows RIF resistance, as well as resistance to FQs. However, because the study did not provide drug susceptibility test results for bedaquiline and linezolid, accurately classifying the bacteria as XDR was challenging.

Genetic mutations associated with drug-resistant TB showed patterns similar to those reported in previous studies [50]. However, some mutations exhibit slightly different effects on drug resistance.

Several disputed mutations that are susceptible to DST but are known to affect treatment outcomes were detected in the present study. Excluding Leu533Pro and Asp516Tyr, the disputed mutations showed high MIC for RIF in the absence of accompanying mutations, and Asp516Tyr-mutant strains were divided into two groups based on the MIC for RIF: five isolates of 2 µg/mL or lower and four isolates of 8 µg/mL or higher. According to the WHO mutation catalogue, the positive predictive value of mutation (PPV) for antibiotic resistance was 69.5% for Leu533Pro and 78.6% for Asp516Tyr. In the case of Ser531Leu, the PPV was 98.9%, and the results of the present study also confirmed resistance, with an MIC of 8 µg/mL or higher. Thus, the contributions of other factors, such as efflux pumps, are expected. As data accumulate in the future, the predictive chart will be able to produce more accurate results.

The mutations were concentrated in His codon 526 and Asp codon 516. In contrast to previous studies, codon 516 exhibited more mutations than codon 526. The detection rate for codon 516 (19.1%) in the present study was more than twice that reported in China and Taiwan (less than 10%) [52,53].

Mutations in *ahpC* are associated with mutations in the *katG* gene. However, in the present study, among the eight isolates of *ahpC* mutants, only one exhibited co-occurrence of a mutation in *katG*. Although C(-52)A, C(-52)T, and ins(-88)GT of *ahpC* have not been reported elsewhere and can be considered natural polymorphisms, G(-48)A, G(-51)A, and C(-81)T are widely recognized mutations that occur in conjunction with *katG* mutations. However, it is challenging to determine whether mutations in *ahpC* indeed affect INH resistance, and it is not yet widely accepted that the *ahpC* mutation alone is involved in INH resistance in *M. tuberculosis* [54].

Met306Ile mutations in *embB* have also been identified in susceptible strains [55,56]. In the present study, 12 isolates of the 26 Met306Ile mutants were in the susceptible range for EMB, but all had an MIC of 4 µg/mL, which was near the critical concentration; therefore, this mutation did not affect resistance but was inferred to decrease susceptibility.

The findings of the present study on resistance to FQ and some *gyrB* mutations differed from the results of other studies. The MIC according to the Arg485Cys mutation of *gyrB* was 4 µg/mL in the present study, albeit with little difference from the results of studies by Farhat et al. (2 µg/mL) and Malik et al. (1 μg/mL) [43,46]. In addition, the Glu540Asp mutation induced slightly higher resistance at levels of 1–2 µg/mL in the present study than at 0.5 µg/mL in the two other abovementioned studies. The Glu540Asp mutants had an MIC for MFX that was 2–4 µg/mL higher than that for OFX. Additionally, in the two studies above, the MIC for MFX was confirmed to be 2 µg/mL, and the mutation had a greater effect on MFX than OFX, unlike other variations. The Ser486Phe mutation was the first clinical isolate identified in the present study, whereas the Asn538Asp and Thr539Asn mutations induced the same level of resistance reported in other studies.

In the present study, the percentages of isolates that did not have mutations related to anti-TB drug resistance were 3.3%, 9.8%, 7.4%, 9.4%, and 7.9% for RIF, INH, FQ, EMB, and AG, respectively. This is because of the contribution of other factors, such as heteroresistance and efflux pumps. However, more than 90% of the resistance mechanisms against major anti-TB drugs could be identified using only the genes identified in the present study.

A secondary aim of this study was to investigate whether there has been an MDR-TB epidemic caused by a specific strain in South Korea. Examination of the genetic relationships of the resistant isolates revealed that all cluster sizes were less than six isolates, and even in the same cluster; except for the *rpoC* mutants, the drug susceptibility patterns were different. Hence, there is no evidence of large-scale epidemics caused by specifically resistant strains. However, the five types of *rpoC* mutations formed clusters consisting of two to five isolates, showing a higher frequency of genetic cluster formation than other resistant strains. This trend has also been reported in other studies [57,58]. Therefore, it is presumed that *rpoC* mutants caused by the *rpoB* mutations are more advantageous in propagation than RIF-resistant strains without the mutation, owing to the compensatory effect of *rpoC* mutations on the fitness cost of *rpoB* mutations.

The recent application of NGS-based DST for TB in clinical settings provides evidence supporting the utility of genotype-based DST. The results obtained using this method will serve as a basis for understanding the effects of specific mutations on drug resistance. Analyzing the effects of each mutation on resistance is expected to further enhance the effectiveness of genetics-based DST.

## 4. Materials and Methods

### 4.1. Mycobacterium tuberculosis Culture and DNA Extraction

A total of 215 *M. tuberculosis* strains were selected from the strains isolated from patients with TB in the Masan National Hospital and the public health network in Korea. Clinical isolates were collected between October 2009 and December 2014. No duplicate isolates of the same origin were detected. Strains were incubated in 10 mL 7H9 broth containing 0.05% (*v*/*v*) Tween 80 and 0.2% (*v*/*v*) glycerol at 37 °C with shaking at 150 rpm. Harvested bacterial cells were allowed to stand for 15 min. The precipitated bacterial cells were transferred to a 1.5 mL microcentrifuge tube in 1 mL volume. The bacterial cells were centrifuged for 2 min at 13,000 rpm, and the supernatant was discarded. DNA was extracted using the InstaGene™ Matrix (Bio-Rad, Hercules, CA, USA) according to the manufacturer’s instructions.

### 4.2. Resistance Gene Sequencing

PCR amplification was performed using the AmpliTaq Gold 360 Master Mix (Applied Biosystems, Foster City, CA, USA). The thermal cycling protocol for *rpoB*, *inhA*, *ahpC*, *gyrA*, *gyrB*, *embB*, *rpsL*, and *rrs* was as follows: predenaturation at 94 °C for 5 min; 25 cycles at 94 °C for 30 s, 58 °C for 30 s, 72 °C for 1 min; and a final extension step at 72 °C for 10 min. The thermal cycling protocol for *katG* was as follows: predenaturation at 94 °C for 5 min; 25 cycles at 94 °C for 30 s, 62 °C for 30 s, 72 °C for 1 min; and a final extension step at 72 °C for 10 min. The thermal cycling protocol for *eis* was as follows: predenaturation at 94 °C for 5 min; 25 cycles at 94 °C for 30 s, 50 °C for 30 s, 72 °C for 1 min; and a final extension step at 72 °C for 10 min. Genomic DNA from the *M. tuberculosis* isolates and reference strains was subjected to PCR. The primer sets used for the amplification and sequencing of resistance-related genes are shown in Appendix A. The PCR amplification products of each gene were sequenced using the same primers as those used by Biofact Co. (Biofact Co. Deajeon, Republic of Korea).

The sequences were analyzed using CLC Main Workbench 6 (CLC bio, Prismet, Aarhus, Denmark). Gene polymorphisms were identified by alignment with the reference strain *M. tuberculosis* H37Rv (GenBank accession no. NC_000962).

### 4.3. Spoligotyping

Spoligotyping is a PCR-based method performed using a spoligotyping kit (Mapmygenome, Hyderabad, India), which was performed according to the manufacturer’s instructions. Genomic DNA was used at a concentration of 100 ng/µL. The spoligotypes were assigned a Spoligo International Type (SIT) and analyzed using SITVITWEB (http://www.pasteur-guadeloupe.fr.8081/SITVIT2_ONLINE/, accessed on 20 June 2020). The SITs were analyzed to similar values using BioNumerics 7.1 (Applied Maths, Sint-Martens-Latem, Belgium).

### 4.4. 24 Locus MIRU-VNTR

The MIRU-VNTR assay was used to describe the triplex PCR-based 24-locus MIRU-VNTR according to Supply et al. [59]. In the present study, we used six sets of quadruplex PCR methods established by Jeon et al. [60]. The amplicons were analyzed using a Genetic Analyzer 3500xL (Applied Biosystems, Waltham, MA, USA). The MIRU-VNTR types are referred to as MV types in this paper. MIRU-VNTR results were analyzed using the BioNumerics 7.1 software (Applied Maths, Sint-Martens-Latem, Belgium).

### 4.5. Antibiotic Susceptibility Testing

*M. tuberculosis* MIC was determined on a Sensititre™ MYCOTBI MIC plate (Trek Diagnostic Systems, Inc., Cleveland, OH, USA) because it is reasonable, rapid, and simple to perform [61,62].

Strains were incubated in 10 mL 7H9 broth containing 0.05% Tween 80 and 0.2% glycerol at 37 °C with shaking at 150 rpm. Harvested bacterial cells were allowed to stand for 15 min. MICs were calculated according to the manufacturer’s protocol. The antibiotics used were OFX, moxifloxacin (MXF), RIF, AMK, STR, rifabutin (RFB), para-aminosalicylic acid (PAS), ethionamide (ETH), cycloserine (CYC), INH, KAN, and EMB. The MIC results were read according to the guidelines of the Clinical Laboratory Standards Institute (CLSI).

## Figures and Tables

**Figure 1 antibiotics-12-01324-f001:**
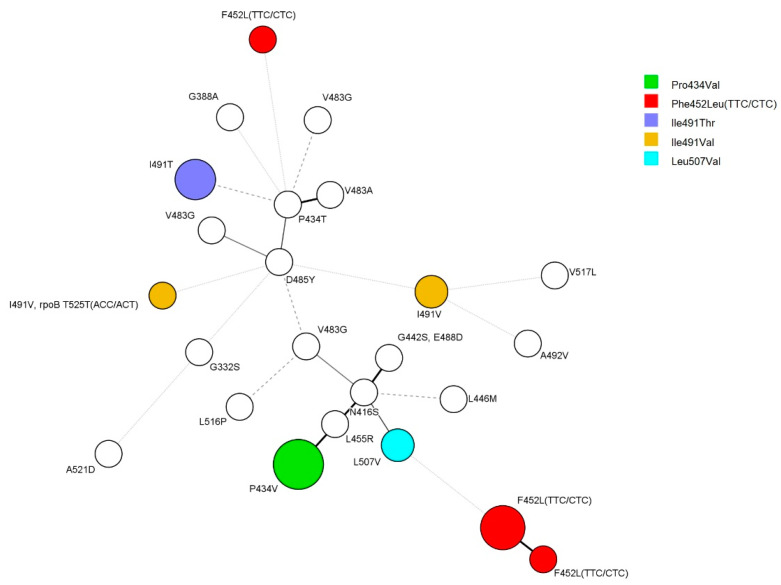
Minimum spanning tree (MST) based on 24-locus MIRU-VNTR. A total of 34 strains were analyzed for *rpoC* mutations. Except for the two strains with Ile491Val and Phe452Leu, each color represents the same mutation. The size of the circles indicates the number of isolates: five strains, Pro434Val; six strains, Phe452Leu; two strains, Leu507Val; two strains, Ile491Val; three strains, Ile491Thr.

**Table 1 antibiotics-12-01324-t001:** Detection of point mutations in *rpoB* for rifampicin MIC in 152 *M. tuberculosis* isolates.

Type of Mutations	Additional Mutation	MIC Range for Rifampicin (μg/mL)	No. of Isolates
Gln513Lys	-	>16	2
Gln513Leu	-	>16	1
513Del	-	16	1
Asp516Gly	rpoB: Leu533Pro rpoB: Leu511Pro	>16	3
Asp516Asn	rpoB: His526Asn	>16	1
Asp516Val	-	16–>16	13
Asp516Tyr	-	0.5–>16	9
	rpoB: Met515Ile	16	1
Ser522Leu	-	16–>16	2
His526Cys	-	>16	1
His526Asp	-	>16	7
His526Gly	-	>16	1
His526Leu	-	8–>16	3
His526Gln	-	>16	1
His526Arg	-	>16	4
His526Tyr	-	>16	7
Ser531Leu	-	8–>16	83
	rpoB: Thr480Ile rpoB: Ins CCG TTC GGG TTC ATC GAA	>16	2
Ser531Trp	-	>16	1
Leu533Pro	-	1–16	7
Ile572Phe	-	2–4	2

**Table 2 antibiotics-12-01324-t002:** Detection of point mutations in *katG*, *inhA*, and *ahpC* for isoniazid MIC in 177 *M. tuberculosis* isolates.

Genes	Type of Mutations	Additional Mutation	MIC Range for Isoniazid (μg/mL)	No. of Isolates
*katG*	Arg249Leu	-	>4	1
	Glu289Gly	inhA: C(-15)T	0.25	1
	Gln295Pro	inhA: G(-17)T	>4	1
	Leu298Ser	-	0.5	2
	Ser315Asn	-	>4	1
	Ser315Thr	-	0.5–>4	86
		inhA: G(-9)A, C(-15)T	>4	8
		katG: Ala243Asp	2	1
		inhA: C(-15)T ahpC: G(-48)A	>4	1
	Ile317Thr	-	0.25	1
	Ala373Cys	inhA: C(-15)T	4	1
	Thr380Ile	inhA: C(-15)T	0.5–2	2
	Trp412stop	-	>4	1
	Gln439Leu	inhA: C(-15)T	1	1
	Ile456Ans	inhA: C(-15)T	1	1
*inhA*	T(-8)A	-	1	1
	T(-8)C	-	0.25–2	4
	G(-9)A	katG: Ser315Thr	>4	1
	C(-15)T	-	0.12–>4	35
		katG: Glu289Gly, Ser315Thr, Ala373Cys, Thr380Ile, Gln439Leu, Ile456Asn	0.25–>4	13
		ahpC: G(-51)A, C(-52)A, del(-79)CA	0.25–>4	3
		katG: Ser315Thr, ahpC: G(-48)A	>4	1
	G(-17)T	katG: Gln295Pro	>4	1
*ahpC*	G(-48)A	-	4	1
		inhA: C(-15)T, katG: Ser315Thr	>4	1
	G(-51)A	inhA: C(-15)T	>4	1
	C(-52)A	-	2	1
		inhA: C(-15)T	0.25	1
	C(-52)T	-	>4	1
	del(-79) CA	inhA: C(-15)T	1	1
	C(-81)T	-	1	1
	ins(-88) GT	-	>4	1

**Table 3 antibiotics-12-01324-t003:** Detection of point mutations in *embB* for ethambutol MIC in 91 *M. tuberculosis* isolates.

Type of Mutations	Additional Mutation	MIC Range for Ethambutol (μg/mL)	No. of Isolates
Ser297Ala	-	2	1
Met306Ile	-	4–32	23
	Asp328Gly, Gly406Asp	8–16	3
Met306Leu	-	16	1
Met306Val	-	4–32	30
Tyr319Asp	-	8	1
Tyr319Asn	Asp354Ala	4	1
Tyr319Ser	-	8	1
Asp354Ala	-	32	1
Asn399Thr	-	4–32	2
Pro404Ser	-	4	1
Gly406Ala	-	4–32	3
Gly406Asp	-	4–16	5
Gly406Ser	-	2	1
Val493Ala	-	1	1
Gln497Lys	-	8–16	3
Gln497Pro	-	4–8	2
Gln497Arg	-	4–32	10
Glu504Asp	-	8	1

**Table 4 antibiotics-12-01324-t004:** Detection of point mutations in *gyrA* and *gyrB* for ofloxacin MIC in 79 and 25 *M. tuberculosis* isolates.

Genes	Type of Mutations	Additional Mutation	MIC Range for Ofloxacin (μg/mL)	No. of Isolates
*gyrA*	His70Arg	-	4	1
		gyrB: Arg485Cys, Thr539Pro	>32	1
	Ala90Val	-	4–>32	15
		gyrA: Asp94Asn, Asp94Ala	>32	2
		gyrB: Arg485Cys, Thr539Pro	>32	2
	Ser91Pro	-	4–16	6
	Asp94Ala	-	4–16	5
		gyrB: Asp500His, Asp500Asn, Thr539Pro	16–>32	3
	Asp94Gly	-	8–>32	25
		gyrB: Asp500Val, Thr539Asn, Gly551Arg	16–>32	6
	Asp94His	-	16	1
		gyrB: Ser486Tyr	>32	1
	Asp94Asn	-	16–32	6
		gyrB: Arg485Leu, Asp500Asn	32–>32	2
	Asp94Tyr	-	8–16	3
*gyrB*	Arg485Cys	-	4	1
		gyrA: Ala90Val	>32	1
		gyrA: His70Arg, gyrB: Thr539Pro	>32	1
	Arg485Leu	gyrA: Asp94His	32	1
	Ser486Phe	-	2–8	3
	Ser486Tyr	gyrA: Asp94His	>32	1
	Asp500His	gyrA: Asp94Ala	>32	1
	Asp500Asn	-	4–8	2
		gyrA: Ala90Val, Asp94Asn, Asp94Ala	>32	3
	Asp500Val	gyrA: Asp94Gly	>32	1
	Asn538Asp	-	4	1
	Thr539Asn	-	4	1
		gyrA: Asp94Gly	>32	1
	Thr539Pro	gyrA: Asp94Ala	16	1
	Glu540Asp	-	1–2	2
	Gly551Arg	gyrA: Asp94Gly	16–32	4

**Table 5 antibiotics-12-01324-t005:** Number of point mutations in *gyrA* and *gyrB*.

No. of Mutated Codons in		*gyrA*			
		0	1	2	Total
*gyrB*	0	78	63	3	144
	1	11	14	-	25
	2	-	1	-	1
	Total	89	78	3	170

**Table 6 antibiotics-12-01324-t006:** Detection of point mutations in *rrs* and *eis* for amikacin MIC in 35 and 10 *M. tuberculosis* isolates, respectively.

Genes	Type of Mutations	Additional Mutation	MIC Range for Amikacin (μg/mL)	No. of Isolates
*rrs*	A514C	-	0.25	1
		eis: G(-37)T	1	1
	A514G	eis: G(-37)T	0.25	1
	C517T	-	0.25–2	4
	A907T	rrs: A908T	0.5	1
	A908T	-	<0.12	1
		rrs: A907T	0.5	1
	A1337C	-	1	1
	A1401G	-	>16	24
	C1402T	-	1	1
*eis*	G(-10)A	-	0.5–1	2
	C(-14)T	-	1	1
	G(-37)T	-	0.5–>16	5
		rrs: A514G, A514C	0.25–1	2

**Table 7 antibiotics-12-01324-t007:** Detection of point mutations in *rpsL* and *rrs* for streptomycin MIC in 38 and 35 *M. tuberculosis* isolates, respectively.

Genes	Type of Mutations	Additional Mutation	MIC Range for Streptomycin (μg/mL)	No. of Isolates
*rpsL*	Lys43Met	-	>32	2
	Lys43Arg	-	>32	22
		rrs: A1401G	>32	5
	Arg86Pro	-	2	1
	Lys88Gln	rrs: A1401G	4	1
	Lys88Arg	-	8–>32	7
*rrs*	A514C	-	2–8	2
	A514G	-	0.5	1
	C517T	-	8–>32	4
	A907T	rrs: A908T	>32	1
	A908T	rrs: A907T	>32	2
	A1337C	-	1	1
	A1401G	-	<0.25–16	18
		rpsL: Lys43Arg, Lys88Gln	4–>32	5
		rpsL: Lys43Arg	>32	1
	C1402T	-	1	1

## Data Availability

Data available on request.

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
