# Peer review of "Molecular Analysis of Anti-Tuberculosis Drug Resistance of Mycobacterium tuberculosis Isolated in the Republic of Korea"

_antibiotics, 2023, doi:10.3390/antibiotics12081324_

Round 1

Reviewer 1 Report

Dear authors! Thank you for the important work, this topic does seem relevant and in demand. Nevertheless, I would like to ask questions that concern me in order to improve the research. 11. Can the MIC analysis be used to assert the validity of the variants found? (line 57) 2. Is the combination of mutations in multiple sites assessed? (line 212) 3. How were patient questionnaires taken into account, including important factors such as treatment, age, gender, etc.? (line 286) 4. The list of literature includes an article by Farhat et al. where they performed a statistical analysis specifying the MIC as a continuous variable. Is this feasible in this project as additional statistic al validation?

Author Response

We are appreciated for the careful and kind review. As suggested by the reviewer, we added the detail explanation

Point 1: Can the MIC analysis be used to assert the validity of the variants found? (line 57)

Response 1: In this study, we confirmed an association between mutations in rpoB and MIC of rifampin. It was confirmed that the MIC for rifampin according to the rpoB mutation could be predicted. Conversely, although a MIC for rifampin can predict a range of rpoB variants, it cannot predict specific types of variants.

Point 2: . Is the combination of mutations in multiple sites assessed? (line 212).

Response 2: We appreciate the careful and kind review.

This section presents the results of analyzing the genetic association of strains based on the 24 locus MIRU-VNTR assay. The only characteristic shared by each genotype cluster was the type of mutation in the rpoC gene.

The passage in Figure 2 and the explanation of this part were insufficient; therefore, the text was revised again to help readers understand.

(line 231-233) Except for the specific rpoC mutant mentioned above, there was little correlation with anti-TB drug susceptibility, even within the cluster.

→ It was difficult to see the correlation between each MIRU-VNTR type with the anti-tuberculosis drug susceptibility pattern or the region and time of isolation. However, there was a correlation with the mutation type of the rpoC gene (Figure 2).

Point 3: How were patient questionnaires taken into account, including important factors such as treatment, age, gender, etc.? (line 286)

Response 3: Unfortunately, clinical information could not be used as the study utilized only clinical strains isolated from patients. Information such as isolation area and period for each strain was used for analysis.

Point 4: The list of literature includes an article by Farhat et al. where they performed a statistical analysis specifying the MIC as a continuous variable. Is this feasible in this project as additional statistical validation?

Response 4:

In this study, the result values of two normal MIC tests were compared. If the values matched, the value from the first test was used. I apologize, but additional statistical validation of MIC values is difficult. 

Reviewer 2 Report

Strains of Mycobacterium tuberculosis are studied in many clinics and universities around the world. In the last three years alone, about 7,000 papers on the study of mutations in the genome of this species of bacteria have been published. The introduction proposed by the authors in no way reflects the achievements available today, it is extremely uninformative and does not reflect the novelty of the authors' research. The cited references are relevant to the work, but they do not reflect the research of recent years. A thorough analysis of the mutations that have arisen in certain genes is given, but the functions of these genes are not considered. Analysis of the impact of identified mutations on the secondary structure and 3d model of gene products is not assessed, although this could provide insight into the mechanisms of the identified mutations on resistance to the antibiotics under study.

Author Response

We are appreciated for the careful and kind review. As suggested by the reviewer, we added the detail explanation

Point 1: Strains of Mycobacterium tuberculosis are studied in many clinics and universities around the world. In the last three years alone, about 7,000 papers on the study of mutations in the genome of this species of bacteria have been published. The introduction proposed by the authors in no way reflects the achievements available today, it is extremely uninformative and does not reflect the novelty of the authors' research. The cited references are relevant to the work, but they do not reflect the research of recent years. A thorough analysis of the mutations that have arisen in certain genes is given, but the functions of these genes are not considered. Analysis of the impact of identified mutations on the secondary structure and 3d model of gene products is not assessed, although this could provide insight into the mechanisms of the identified mutations on resistance to the antibiotics under study.

Response 1: We appreciate the careful and kind review.

As suggested by the reviewer, there are many papers related to gene mutation in Mycobacterium tuberculosis.

It is well known that the treatment of tuberculosis takes at least 6 months even for susceptible tuberculosis that is not resistant to anti-tuberculosis drugs; therefore, a quick and accurate diagnosis of resistance is required.

Since phenotype tests such as mic analysis used to diagnose resistance to anti-tuberculosis drugs take at least two months, it is possible to check whether or not mutations in resistance-related genes exist and the types of mutations based on recent NGS, and to determine resistance based on these results. Predictable early diagnosis methods are being developed and applied.

In this diagnosis, it is important to predict the degree of resistance according to the type of mutation; therefore, the results of various studies are continuously being collected. This manuscript can provide the information needed to predict the type of mutation in specific genes and the degree of resistance accordingly.

Reviewer 3 Report

The manuscript of Se-Mi Jeon and co-authors (Molecular analysis of the anti-tuberculosis drug resistance of Mycobacterium tuberculosis isolated in the Republic of Korea) analysed the occurrence of antibiotic-resistant strains of M. tuberculosis  in South Korea. Here, several genes were sequenced. Attempts have been made to correlate these mutations with the emergence of resistance. These analyses are important in terms of tracking possible epidemics and therefore publication of the manuscript is justified. However, there are several points that should be changed or corrected.

  1. a more detailed explanation of the genes studied is missing. 2. please indicate the corresponding MIC 's of the sensitive wild-type strain. 3. lanes 61-62: without further explanation the sentence (No mutations were found...) makes no sense. 4. lanes 65-66: please check the information on the mutation Ser531Leu with the information in Table 1, these do not match. 5. figure 1 can be deleted, as no more information is given here that is not already in table 4. 6. lanes 144-145: why were not all strains analyzed? What criteria were used for selection? 7. lanes 328-330: how often were the analyses for MIC carried out? How high were the standard deviations?

Author Response

We are appreciated for the careful and kind review. As suggested by the reviewer, we added the detail explanation

Point 1: a more detailed explanation of the genes studied is missing.

Response 1: We appreciate the careful and kind review. As suggested by the reviewer, I have added the description of the genes under study.(lines 33-66)

Point 2: please indicate the corresponding MIC 's of the sensitive wild-type strain.

Response 2: Normally, the susceptible standard strain, Mycobacterium tuberculosis ATCC27294 H37Rv, was tested at every MIC test, and the results are as follows

This result was created as a supplementary table because it was difficult to match the format with the tables in the text.

Point 3: lanes 61-62: without further explanation the sentence (No mutations were found...) makes no sense

Response 3: We apologize for our insufficient explanation. We added a detailed explanation in the lines 95-97.

Point 4: lanes 65-66: please check the information on the mutation Ser531Leu with the information in Table 1, these do not match.

Response 4: Thank you for pointing out the careful error. The text was corrected to 16 μg/ml (line 100)

Point 5: figure 1 can be deleted, as no more information is given here that is not already in table 4.

Response 5: I agree with the reviewer's point. Figure 1 was removed.

Point 6: lanes 144-145: why were not all strains analyzed? What criteria were used for selection?

Response 6: We appreciate the reviewer's interest.

First, mutations in the eis gene were checked for all AG-resistant strains. Subsequently, mutations were also checked for susceptible strains. At the time, laboratory circumstances prevented further experiments at 36 isolates. Therefore, internal discussions were held on whether to include the mutation results for the eis gene in this manuscript. Although there are no special selection criteria for AG-susceptible strains, it was judged that the data was useful and included it.

Point 7: lanes 328-330: how often were the analyses for MIC carried out? How high were the standard deviations?

Response 7: In this study, after running a normal MIC test twice, if the result values matched, the value was selected and used. I apologize, but additional statistical validation of MIC is difficult.

Round 2

Reviewer 2 Report

This topic has been widely studied, and especially the mutations mentioned in the article, in the last two years. It is therefore extremely strange that the authors do not actually do a review of the published information. If this had been done, the novelty and significance of the findings would have been obvious. As the article is presented, it looks like a report on the observations made. Obviously, the authors should significantly improve the text of Introduction and Discussion parts of the article.

Author Response

Response to Reviewer 2 Comments

We are appreciated for the careful and kind review. As suggested by the reviewer, we added the detail explanation

Point: This topic has been widely studied, and especially the mutations mentioned in the article, in the last two years. It is therefore extremely strange that the authors do not actually do a review of the published information. If this had been done, the novelty and significance of the findings would have been obvious. As the article is presented, it looks like a report on the observations made. Obviously, the authors should significantly improve the text of Introduction and Discussion parts of the article.

Response: I agree with the reviewer's opinion. Over the last two years, improved studies using various methods, such as DST analysis using WGS, have been published. Therefore, the papers cited have been reinforced, and the introduction and discussion have been modified accordingly.